# REFLECTION TRIGGER: LATENT SELF-CORRECTION FOR QUESTION ANSWERING BY STEERING VECTOR INJECTION

## ABSTRACT

Large language models (LLMs) excel at reasoning tasks, but achieving stable reflective reasoning remains a challenge. Existing techniques, such as prompt engineering and multi-turn prompting, often lead to over-reflection, unstable outputs, and heavy reliance on manually designed prompts. In response to these limitations, we propose *Reflection Trigger*, a novel vector-based mechanism that dynamically injects the reflection vector into LLMs during inference without modifying model parameters. These vectors, based on latent semantic representations, are trained to encode reflection signals. By training a module to generate input-specific reflection vectors, our method provides a controllable and stable mechanism to adjust the model's internal reflection tendencies. Experiments on biomedical and commonsense benchmarks demonstrate that the Reflection Trigger improves reasoning accuracy and reduces over-reflection. These results suggest that the Reflection Trigger enhances the stability of LLM reasoning and show that reflective reasoning can be treated as a learnable and controllable capability.

## 1 INTRODUCTION

Large language models (LLMs) have demonstrated remarkable performance across a wide range of NLP tasks (Brown et al., 2020; Touvron et al., 2023; Bang et al., 2023), including data analysis, reading comprehension, logical reasoning, and open-domain question answering (Zhu et al., 2024; Wang et al., 2024). However, LLMs often generate responses that appear plausible but are factually incorrect. Such errors frequently arise from intuitive reasoning failures, where the model relies on superficial linguistic patterns rather than deeper logical inference or self-correction (Hagendorff et al., 2022).

To improve the quality of reasoning, the Chain-of-Thought (CoT) prompting strategy has been proposed, enabling models to generate intermediate reasoning steps (Wei et al., 2022). CoT has proven effective in tasks requiring multi-step logical inference, mathematical problem solving, and commonsense reasoning (Kojima et al., 2022). However, subsequent studies have shown that even with CoT, models still suffer from flawed reasoning, such as logical inconsistencies, calculation mistakes, and hallucinated content. This suggests that although CoT enhances the structure of reasoning, it does not effectively resolve the problem of intuitive misjudgment (Ji et al., 2023a; Hadi et al., 2023).

To further address such errors, recent research has proposed the approach of reflection (Ji et al., 2023b), aimed at enabling models to mimic the human ability to rethink their responses after making errors (Renze & Guven, 2024; Shah et al., 2025). Current mainstream reflection strategies mostly rely on prompt-based control, using prompts such as "Please check if the previous response was wrong" (Renze & Guven, 2024) or "Wait," (Shah et al., 2025; Muennighoff et al., 2025) guide the model toward self-reflection and correction.

However, prompt-based strategies have several limitations:

1. **Lack of adjustable reflection intensity**: Prompt-based methods are binary, where the model either reflects or does not, without dynamic control.

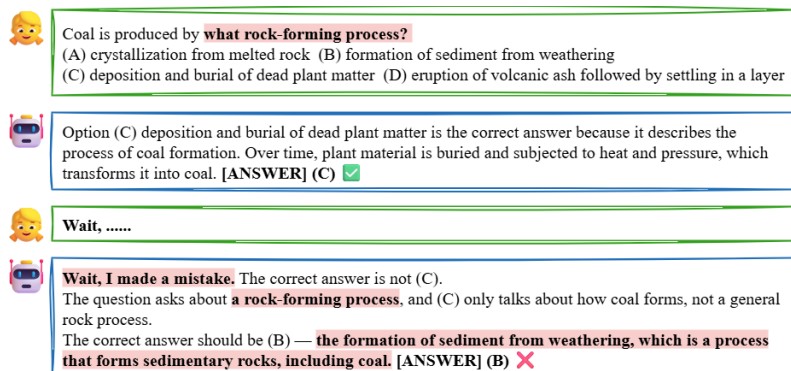

Figure 1: This example illustrates a failure case where the LLM's reflective process leads to over-reflection.

2. **Increased latency and behavioral instability**: Multi-turn prompting significantly increases inference latency and computational resource consumption, and may also lead to inconsistent model behavior across different prompts.

3. **Risk of over-reflection**: *Over-reflection* occurs when the model "overthinks" and unnecessarily revises an initially answer into an incorrect one, leading to reduced accuracy. For example, as shown in Figure 1, the model initially selected the correct answer to a coal-formation question, but after reflection, it misinterpreted the term *rock-forming process* and changed its response to an incorrect option. This case demonstrates that if the reflection mechanism is not properly controlled, reflection itself may lead to degraded rather than improved reasoning performance.

To address the limitations, we propose a vector-guided reflection control mechanism, called the *Reflection Trigger*. As shown in Figure 2, this mechanism injects a learned vector into the intermediate layers of the model during inference, allowing for dynamic adjustment of reflective tendencies across different questions. Unlike prompt-based methods, Reflection Trigger s an internal mechanism, capable of effectively guiding model reflection. Moreover, it achieves this without requiring additional inference rounds or parameter fine-tuning.

To summarize, the primary contributions of our study are:

1. We propose a vector-based method for controlling reflective reasoning in LLMs.

2. Our method requires no fine-tuning LLMs and can dynamically adjust the model's reflection tendencies based on each input question.

3. Demonstrates that the proposed Reflection Trigger mechanism effectively improves model accuracy while reducing over-reflection errors, achieving robust performance in both commonsense and biomedical reasoning tasks.

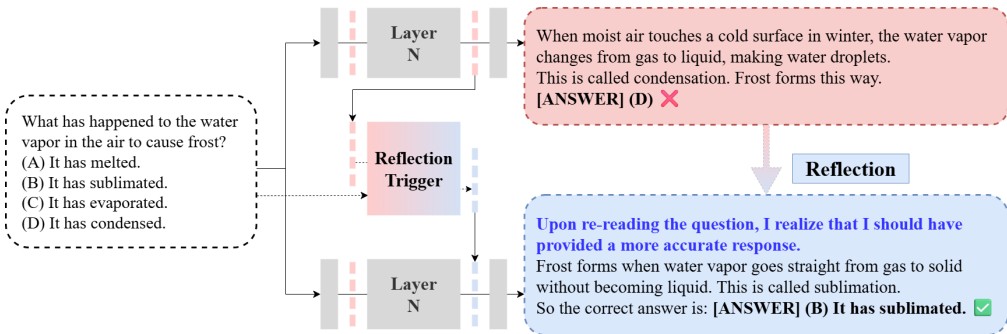

Figure 2: Our proposed method —Reflection Trigger.

## 2 RELATED WORK

**Self-Reflection in Language Models.** Self-reflection has recently become an effective mechanism for enhancing the reasoning ability of language models (Ji et al., 2023b; Wan et al., 2025). Prior work has explored prompting models to generate intermediate reflections, with methods such as Reflexion (Shinn et al., 2023), SELF-REFINE (Madaan et al., 2023), and ReAct (Yao et al., 2023) showing improvements in problem solving and planning. However, most approaches rely on explicit prompts and treat reflection mainly as error correction, overlooking scenarios where the initial answer is already correct. When reflection is forced in such cases, the model may cause the model to doubt correct reasoning and change answers unnecessarily. Moreover, these approaches treat reflection as a binary switch, ignoring the need for context-sensitive and dynamically regulated reflection. Therefore, we propose *treating reflection as a semantic direction vector instead of a single prompt phrase.*

**Steering via Latent Representations.** Beyond prompt engineering and fine-tuning, recent studies have explored controlling LLM behavior through latent representations. Early methods such as Plug-and-Play Language Models (PPLM) (Dathathri et al., 2019) achieved sentiment control by introducing activation shifts during the generation process. LoRA (Low-Rank Adaptation) (Hu et al., 2022), on the other hand, proposed a parameter-efficient adaptation method, where trainable low-rank update modules are added to specific weight matrices, enabling fine-tuning effects with minimal parameter modification. More recently, activation engineering techniques have been proposed to control model outputs by directly modifying activations in intermediate layers, without modifying the model's original weights (Hernandez et al., 2023; Panickssery et al., 2023; Javaid et al., 2024; Stolfo et al., 2024). Latent steering vectors have been identified to guide outputs toward specific target sentences, but this approach requires costly per-sentence optimization (Subramani et al., 2022). The Activation Addition (ActAdd) method constructs steering vectors from differences in activations between two semantically contrasting prompts (Turner et al., 2023). However, this method suffers from inconsistent performance across different behaviors and task settings.

Motivated by these limitations, we introduce Reflection Trigger, a vector-based steering mechanism specifically designed for reflective reasoning. Unlike prior methods focused on style or knowledge control, our approach directly predicts a latent vector based on the input question to control the reflective tendency of the model's reasoning. This design enables controllable reflective behavior without prompt engineering or fine-tuning, combining efficiency with stability.

## 3 METHODOLOGY

In this section, we propose the Reflection Trigger, a vector-based mechanism designed to guide large language models (LLMs) toward reflective reasoning without relying on prompt engineering or parameter fine-tuning. The core idea is to inject a learned latent control vector into the intermediate representations of the model, which enables stable and controllable induction of more reflective reasoning behaviors.

### 3.1 OVERALL FRAMEWORK

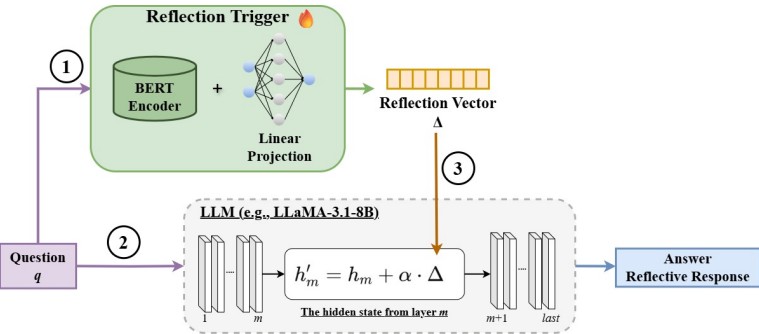

Figure 3: Overall Framework of Reflection Trigger.

As shown in Figure 3, our framework consists of three main components:

1. **Reflection Trigger:** A pretrained BERT encoder (Devlin et al., 2019) with linear projection generates a latent vector from the semantic embedding of the input question. This vector captures the tendency for reflective reasoning.

2. **LLM Reasoning:** The input question is simultaneously fed into a frozen LLM to obtain hidden states from an intermediate layer $m$.

3. **Vector Injection:** The predicted *reflection vector* $\Delta$ is injected into the hidden representation as $h'_m = h_m + \alpha \cdot \Delta$, where $\alpha$ is a tunable coefficient controlling reflection intensity. This adjustment influences the subsequent reasoning and output generation of the model.

Through this mechanism, the model can revise initial reasoning internally within a single forward pass, improving stability.

## 3.2 REFLECTION TRIGGER

The goal of the Reflection Trigger is to predict a reflection vector that encodes reflection tendencies for a given input question. This vector is injected into the intermediate layers of the LLM during inference, enabling controllable adjustment of reasoning behavior without modifying model parameters.

### 3.2.1 TRAINING DATA CONSTRUCTION

To supervise the learning of reflection behavior, we construct reflection pairs for each input question $q$, as shown in Figure 4.

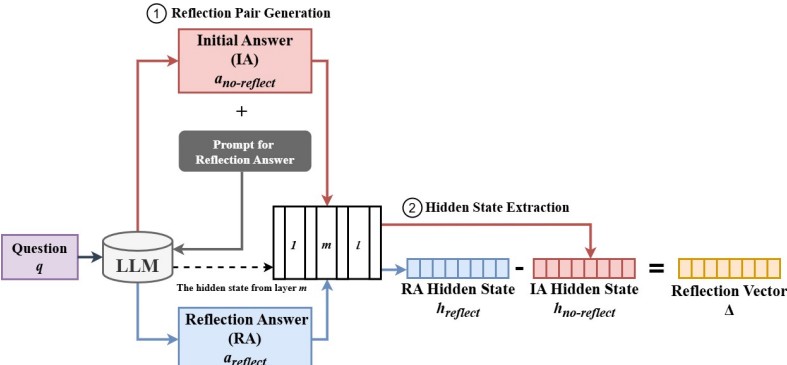

Figure 4: Training data construction pipeline. The pipeline consists of two main stages. (1) **Reflection Pair Generation:** given a question $q$, the LLM first produces an initial answer ($IA$) without reflection, and then, under a reflection prompt, generates a reflection answer ($RA$). (2) **Hidden State Extraction:** hidden representations corresponding to both $IA$ and $RA$ are extracted from the same intermediate layer of the LLM. The difference between the $RA$ hidden state $h_{\text{reflect}}$ and the $IA$ hidden state $h_{\text{no-reflect}}$ is computed to form the reflection vector $\Delta$, which encodes the semantic direction of reflection. This vector serves as the training target for our Reflection Trigger model.

A reflection pair consists of:

- **Initial Answer** ($IA$): $a_{no-reflect}$, generated without reflection prompts, representing the model's intuitive reasoning.

- **Reflection Answer** ($RA$): $a_{reflect}$, generated with a reflection-inducing prompt, representing the model's reflective reasoning.

For each reflection pair, hidden states are extracted from the same intermediate layer of the LLM. The semantic difference between $RA$ and $IA$ is defined as the reflection vector:

$$\Delta = h_{\text{reflect}} - h_{\text{no-reflect}}, \tag{1}$$

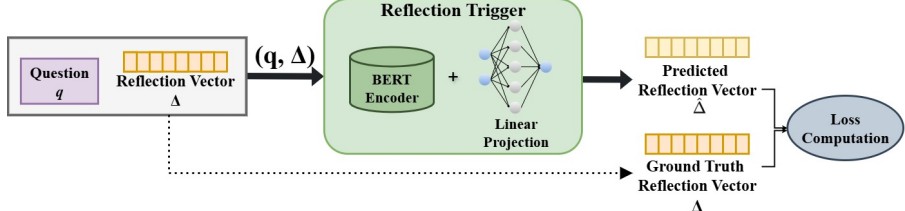

Figure 5: Supervised training process of the Reflection Trigger. Given a question and its reflection vector, the BERT encoder and linear projection are trained to predict the reflection vector $\Delta$, which serves as ground-truth supervision. This process enables the model to learn input-specific reflection control.

Where $h_{\text{reflect}}$ and $h_{\text{no-reflect}}$ are the hidden states associated with $RA$ and $IA$. This vector represents the transition from intuitive to reflective reasoning and serves as the ground-truth supervision signal for training the Reflection Trigger. We refer to this difference vector as the reflection vector, which can be interpreted as a directional guide in semantic space. While inspired by the previously proposed steering vector methods, our application extends from style or knowledge control to reasoning style guidance. The prompt used for training data construction is provided in A.3.1.

### 3.2.2 MODEL TRAINING

The objective of model training is to enable the Reflection Trigger to predict an appropriate reflection vector $\Delta$ directly from an input question $q$. To achieve this, we adopt a supervised learning framework based on a pretrained BERT encoder.

The model architecture is as follows (Figure 5):

- A pretrained BERT encoder maps the input question $q$ into a semantic representation. Its relatively lightweight architecture enables efficient computation with low inference overhead, making it suitable for generating steering vectors without incurring significant latency.
- This representation is then passed through a linear projection layer, which maps it into the same dimensional space as the intermediate hidden state of the target language model.
- The objective of the model training is to minimize the distance between the predicted reflection vector $\hat{\Delta}$ and the ground-truth vector $\Delta$, which is obtained from the data construction process.

### 3.3 LLM REASONING AND INJECTION

During inference, the predicted reflection vector is injected into the hidden representation of the LLM at layer $m$. This modulation steers the reasoning toward reflective tendencies, enabling the model to correct errors while mitigating over-reflection. Importantly, the procedure requires no parameter updates or fine-tuning. Only adjusts the internal representation through a latent semantic vector.

## 4 EXPERIMENTS

Our experiments are designed to answer the following research questions: **RQ1:** Can Reflection Trigger improve reasoning accuracy across domains? **RQ2:** Does Reflection Trigger reduce over-reflection while enabling effective self-correction? **RQ3:** How does Reflection Trigger compare with prompt-based reflection methods in terms of over-reflection control and efficiency?

### 4.1 EXPERIMENT SETTINGS

**Datasets.** To evaluate the effectiveness and generalizability of the proposed Reflection Trigger, we conduct experiments in two major reasoning domains: commonsense and biomedical reason-

Table 1: Performance comparison.

| Dataset | Vanilla | LoRA | Prompt-based | | Reflection Trigger | |
|---|---|---|---|---|---|---|
| | | | CoT | "Wait," | In-Domain | Cross-Domain |
| *Biomedical Reasoning* | | | | | | |
| MedQA | 55.53 | 58.52 | 60.02 | 46.11 | **64.10** | 63.86 |
| MedMCQA | 53.36 | **59.10** | 55.92 | 50.00 | 56.44 | 57.52 |
| MMLU-Med | 67.58 | - | **75.85** | 49.77 | **75.85** | 74.47 |
| *Commonsense Reasoning* | | | | | | |
| ARC-Challenge | 79.18 | 76.11 | 83.19 | 61.18 | **83.28** | 82.94 |
| CSQA | 73.14 | **81.16** | 74.69 | 65.46 | 72.89 | 71.50 |
| **Average Accuracy (%)** | 65.76 | 68.72 | 69.93 | 54.50 | **70.51** | 70.06 |

ing. These two domains differ significantly in their semantic structure, reasoning complexity, and knowledge-based requirements. For medical reasoning, we use MedQA (Jin et al., 2021), MedM-CQA (Pal et al., 2022), and MMLU-Med (Hendrycks et al., 2020). For commonsense reasoning, we use ARC-Challenge (Clark et al., 2018) and CommonsenseQA (CSQA) (Talmor et al., 2019). Details of the data filtering procedure for reflection training are provided in A.3.1.

**Baselines.** We compare our proposed Reflection Trigger method with three groups of baselines, all using the same backbone model: *Llama3.1-Instruct-8B* (Dubey et al., 2024). 1) The **vanilla** baseline refers to the original *Llama3.1-Instruct-8B* model without any intervention; 2) **Prompt-based** reflection methods, including Chain-of-Thought (CoT) (Wei et al., 2022), which guides the model to reason step-by-step via explicit prompting, and the "Wait," prompt (Shah et al., 2025; Muennighoff et al., 2025), a multi-turn prompting strategy that introduces a reflective signal to prompt the model to rethink its initial reasoning process and 3) **LoRA (Low-Rank Adaptation)** (Hu et al., 2022) represents a parameter-efficient fine-tuning approach, which requires training a separate LoRA model for each dataset.

**Evaluation Metrics.** We evaluate model performance using accuracy as the primary metric, and additionally provide *reflection rate* and *over-reflection* rate to analyze reflective behavior. These metrics measure both task effectiveness and the stability of reflection. The details of the evaluation metrics are described in A.2.

**Implementation details.** Reflection Trigger uses a BERT encoder (Devlin et al., 2019) to generate reflection vectors from reflection pairs. These vectors are injected into the hidden representations of *LLaMA3.1-Instruct-8B*. Unless otherwise specified, all experiments are conducted with the injection layer set to 16 and the reflection intensity fixed at $\alpha = 1.0$.

## 4.2 RQ1: REFLECTION TRIGGER PERFORMANCE COMPARISON

**Cross-Domain generalization.** To evaluate the generalization capability of the Reflection Trigger, we compare in-domain and cross-domain performance. In this setting, the reflection vector is trained on one task and then applied to a different domain to assess transferability. The results indicate that Reflection Trigger maintains stable accuracy across domains. For example, it achieves 82.94% on ARC-Challenge and 71.50% on CSQA when the reflection vector is trained on biomedical reasoning tasks. These performances are remarkably close to the in-domain results, suggesting that the reflection strategy learned is highly transferable. This further implies that the Reflection Trigger enables broadly applicable reflective reasoning, without the need for task-specific fine-tuning.

**Overall performance.** The Reflection Trigger achieves an average accuracy of 70.51% on five reasoning tasks, outperforming all comparison methods without relying on fine-tuning or prompt engineering. It achieves the best performance on both MedQA and ARC-Challenge, with accuracies of 64.10% and 83.28%. This confirms a latent reflection vector injection effectively steers LLM reasoning.

On the MedMCQA and CSQA tasks, LoRA slightly outperforms Reflection Trigger. This is likely because parameter fine-tuning directly enhances the model's ability to memorize domain-specific

Table 2: Comparison of prompt-based methods on reasoning tasks.

| Task | Methods | Accuracy (%) | Reflection Rate (%) | Over-reflection Rate (%) | Generated Tokens |
|---|---|---|---|---|---|
| **Biomedical Reasoning** | CoT Prompt | 63.93 | - | - | 284.50 |
| | "Wait," Prompt | 48.63 | 27.26 | 33.02 | 289.67 |
| | **Reflection Trigger (Ours)** | **65.46** | **12.68** | **8.48** | **126.92** |
| **Commonsense Reasoning** | CoT Prompt | **78.94** | - | - | 183.71 |
| | "Wait," Prompt | 63.32 | 39.19 | 37.45 | 426.93 |
| | **Reflection Trigger (Ours)** | 78.09 | **15.48** | **7.94** | **73.43** |

knowledge, which is particularly effective on large-scale datasets. Although Reflection Trigger can guide the model to think more reflectively, its improvements may be limited in tasks where the model lacks sufficient prior knowledge. Meanwhile, the "Wait," prompt performs poorly across all tasks, with an average accuracy of only 54.50%. This suggests that although multi-turn prompting provides a straightforward mechanism for reflection, it often makes the model over-doubt its initial answer and degrades performance.

Based on these results, we conclude that Reflection Trigger is stable, requires no modification to the model architecture, and generalizes effectively across reasoning tasks. Reflection should *not be viewed only as error correction, but as a learnable, controllable, and semantically guided reasoning strategy.* Our experiments further show that injecting a latent semantic vector is sufficient to steer a model's reasoning style without relying on prompt-based guidance or parameter modification.

### 4.3 RQ2: COMPARISON WITH PROMPT-BASED REFLECTION METHODS

We compare Reflection Trigger with two representative prompt-based methods: CoT and the "Wait" prompt, across both commonsense and biomedical reasoning tasks. As shown in Table 2, Reflection Trigger achieves higher average accuracy in both domains without the need for designed prompts. Moreover, we examine two important behavioral metrics: *reflection rate* and *over-reflection rate* (Section A.2)), as introduced in Section 4.3. For example, in the commonsense reasoning task, the over-reflection rate of Reflection Trigger is 7.94%, significantly lower than the 37.45% observed with the "Wait," prompt. This indicates that our method effectively mitigates over-reflection, leading to more stable reasoning.

In addition, Reflection Trigger produces significantly shorter outputs compared to the two prompt-based methods. In biomedical reasoning tasks, it generates an average of only 126.92 tokens, compared to 285–290 tokens from CoT and "Wait," prompt. In commonsense reasoning, Reflection Trigger outputs only 73.43 tokens on average, which is significantly fewer than the 426.93 tokens generated by the "Wait," prompt. This reduction in output length suggests that our method offers lower latency and improved efficiency during inference.

### 4.4 RQ3: SENSITIVITY TO INJECTION PARAMETERS AND TRAINING DATA SIZE

**Parameter sensitivity.** To evaluate the robustness of Reflection Trigger with respect to hyperparameters, we conduct experiments on two tasks: MMLU-Med (biomedical reasoning) and CSQA (commonsense reasoning). Specifically, we inject the reflection vector into layers 8, 12, 16, 20, and 24, and vary its strength by setting the injection coefficient to 0.5, 1.0, 2.0, 3.0, or 5.0

As shown in Figure 6, across both commonsense and biomedical tasks, we observe that excessively strong reflection signals consistently degrade performance, especially when injected into shallow layers. On CSQA, accuracy drops to 29.24%, indicating that overly strong injection disrupts the model's semantic distribution, causing semantic distortion and over-reflection. In contrast, injecting the reflection vector into mid-to-upper layers yields more stable results. This is consistent with transformer design, where higher layers are responsible for semantic integration and generate final outputs. Thus, injecting reflection signals into deeper layers can effectively guide reflective behavior while minimizing interference with lower-layer language understanding.

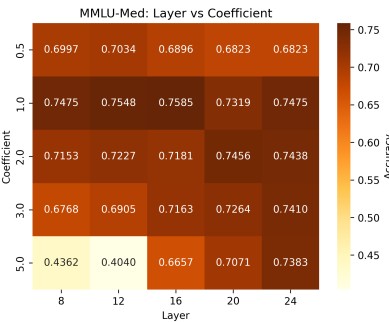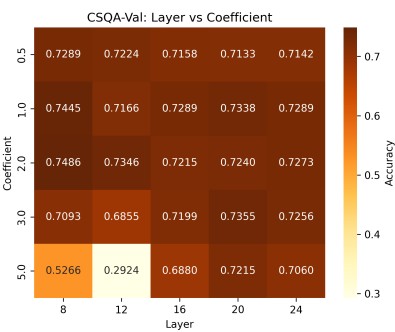

Figure 6: Heatmaps of Reflection Trigger under different injection layers and coefficients. The horizontal axis indicates the injection layer, and the vertical axis shows the scaling coefficient of the reflection vector. Each cell reports the corresponding accuracy, with darker colors representing higher accuracy and lighter colors representing lower accuracy.

Moreover, a coefficient of 1.0 already provides stable reflective guidance. This supports our design principle that both the strength and tendency of reflection are learned automatically during training, without requiring manual tuning of injection intensity. Further analysis of reflection intensity is provided in A.4.1, which shows our method adapts across task difficulties and domains.

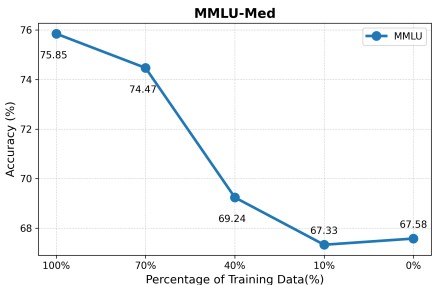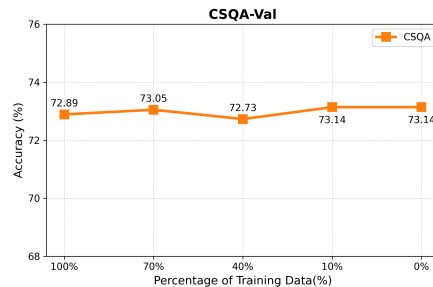

Figure 7: The accuracy of the model varies across different proportions of Reflection Trigger training data on two tasks: MMLU-Med and CSQA. The X-axis represents the percentage of training data used (from 0% to 100%), while the Y-axis shows the corresponding accuracy (%) on each test set.

**Analysis of training data efficiency.** As shown in Figure 7, we further analyze the impact of training data volume on reflective capability. On MMLU-Med, when training data is reduced from 100% to 70%, the accuracy drops only slightly by 1.38 points, indicating that reflective capability remains stable with moderate data reduction. However, when the data are below 40%, performance dropped a lot, showing that inadequate training data makes it hard for the model to learn reflection. In particular, the results at 0% and 10% data are nearly identical, implying that without sufficient reflective samples, the model fails to acquire meaningful reflective reasoning ability.

In contrast, on CSQA, the model's performance remains almost unchanged across all data volumes, maintaining accuracy between 72.89 and 73.14. Even with only 0% or 10% training data, the model achieves comparable results. This indicates that for tasks where the model already has strong prior knowledge, additional reflection training provides minimal performance improvement.

## 5 CASE STUDY

### 5.1 CASE STUDY 1—CONTROLLED REFLECTION PREVENTS OVERTHINKING

As shown in Table 3, this case shows how too much reflection can affect model performance. The vanilla and Reflection Trigger outputs both correctly choose (D), with Reflection Trigger providing a stronger explanation by referencing Newton's first law. However, the "Wait," prompt causes the

model to overthink, doubt its original answer, and wrongly choose (A). This example highlights that while reflection can help, excessive reflection can mislead the model.

## 5.2 Case Study 2—Successful Reflection Case

As shown in Table 4, the model initially selected the wrong answer (D) rubbing alcohol, based on a flawed understanding of how to separate salt from water. However, with the Reflection Trigger applied at a low coefficient ($\alpha = 1.0$), the model successfully revised its answer to (B) heat, correctly identifying evaporation as the key process. This shows that even with minimal intervention, the Reflection Trigger can effectively guide the model to correct its reasoning. By contrast, the 'Wait' prompt requires multi-turn prompting to answer the correct answer, which increases computational overhead and response latency.

Table 3: Case study 1—controlled reflection prevents overthinking.

| Question | A car is rolling on a level street. What can make the car stop moving? (A) a force that pushes straight up (B) a force that pushes straight down (C) a force that pushes the same direction that the car is moving (D) a force that pushes the opposite of the direction that the car is moving |
|---|---|
| Vanilla | In order for the car to stop moving, a force must be applied in the opposite direction of its motion. This is known as the opposite force principle, which is a fundamental concept in physics. [ANSWER](D)[/ANSWER] |
| Reflection Trigger ($\alpha = 1.0$) | A force that pushes in the opposite direction of the car's motion is necessary to stop the car. This is because, according to Newton's first law of motion, an object in motion will continue to move unless acted upon by an external force. [ANSWER](D)[/ANSWER] |
| "Wait," Prompt | Wait, there is another option, a force that pushes straight up is actually opposite to the force that is acting on the car, the force of gravity. So the correct answer is a force that pushes straight up. [ANSWER](A)[/ANSWER] |

Table 4: Case study 2—successful reflection case.

| Question | The best way to separate salt from water is with the use of (A) oil. (B) heat. (C) a magnet. (D) rubbing alcohol. |
|---|---|
| Vanilla | The best way to separate salt from water is with the use of (D) rubbing alcohol. The reasoning behind this is that rubbing alcohol (isopropanol) is a polar solvent that can dissolve salt, allowing it to be separated from water through a process called extraction. When you mix rubbing alcohol with a saltwater solution, the salt will dissolve in the alcohol, leaving the water behind. This process is often used in chemistry labs to separate salt from water. [ANSWER](D)[/ANSWER] |
| Reflection Trigger ($\alpha = 1.0$) | The best way to separate salt from water is by using heat, as it allows for the evaporation of the water, leaving the salt behind. This process is commonly known as desalination. [ANSWER](B)[/ANSWER] |
| "Wait," Prompt | Wait, that's not the right answer. Let me look at the options again. (D) is not correct because rubbing alcohol isn't typically used to separate salt from water, and salt doesn't dissolve better in alcohol than in water. The other options — (A) oil and (C) a magnet — don't work for separating salt from water. On the other hand, (B) heat can be used to evaporate the water, leaving the salt behind. So, the best choice here is (B). [ANSWER](B)[/ANSWER] |

## 6 Conclusion

In this work, we propose the Reflection Trigger, an approach to inducing reflective reasoning in large language models (LLMs) by injecting a learned vector into their hidden representations. This method requires no parameter modification or prompt engineering. Through comprehensive evaluations on both commonsense and biomedical reasoning tasks, our method consistently improves accuracy while maintaining high stability and efficiency. Compared with prompt-based techniques such as Chain-of-Thought (CoT) and "Wait," prompts, Reflection Trigger achieves comparable performance, with significantly fewer generated tokens and lower over-reflection rates.

Further analysis shows that the method is robust across hyperparameters and tasks, indicating that the strength and tendency of reflective reasoning can be learned rather than manually encoded through prompt design. In summary, Reflection Trigger offers a simple yet effective mechanism for influencing model behavior through semantic-level intervention, enabling more controllable and stable reasoning without the need for prompts or model fine-tuning.

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

# A APPENDIX

## A.1 DATASETS

To evaluate the effectiveness of the proposed Reflection Trigger mechanism on diverse reasoning tasks, we conducted experiments in two domains: commonsense and biomedical reasoning. Table 5 summarizes the datasets used for each domain.

To improve the effectiveness of reflective behavior learning, we apply a filtering process to the training data used for the Reflection Trigger module. Following the data construction method described in 3.2.1, we retain only those samples in which the reflection process results in a correct answer, forming the final filtered training set.

These samples are categorized into two types:

- Error Correction (*Wrong → Correct*): The model's initial response is incorrect, but reflection enables it to revise the answer and arrive at the correct solution.

- Reaffirmation (*Correct → Correct*): The model initially gives the correct answer and confirms its reasoning and reaffirms the correct answer through reflection.

Table 5: Dataset statistics for Reflection Trigger training and testing. Raw training data are first generated via reflection pairs, then filtered to retain only cases where reflection leads to a correct answer (*Wrong → Correct* or *Correct → Correct*). MMLU-Med does not provide training data and is used only for evaluation.

| Domain | Dataset | # of Raw Training Data | # of Filtered Training Data | # of Testing Data |
|---|---|---|---|---|
| **Biomedical Reasoning** | MedQA | 10,178 | 4,471 | 1,273 |
| | MedMCQA | 182,822 | 6,012 | 4,183 |
| | MMLU-Med* | - | - | 1,089 |
| **Commonsense Reasoning** | ARC Challenge | 1,119 | 722 | 1,172 |
| | CSQA | 9,741 | 6,042 | 1,221 |

This filtering strategy helps focus the learning on successful reflective reasoning, avoiding the noise introduced by failed or inconsistent samples. Unlike traditional approaches that emphasize only the "wrong-to-right" correction pattern, we argue that reflection should be viewed as a consistent reasoning style, not merely a mechanism for correcting mistakes. Therefore, we also include cases where the model originally got the answer right and confirmed it again during reflection, to help reinforce this stable reflective reasoning style.

## A.2 EVALUATION METRICS

To comprehensively evaluate the performance of the proposed Reflection Trigger method on reasoning tasks, we adopt standard accuracy as the primary metric, and design two additional metrics to analyze the specific impact of the reflection mechanism on the reasoning behavior of the model.

### A.2.1 ACCURACY

The primary metric is accuracy, which measures the proportion of correctly answered questions. Since all models follow a fixed output format, the final selected answer is extracted by identifying the token enclosed in the pattern: `[ANSWER](choice letter)[/ANSWER]`. This formatting ensures consistent detection and enables automated accuracy calculation across different prompting methods.

### A.2.2   REFLECTION RATE

To measure whether reflection actually helps correct the original error, we define the reflection rate, defined as:

$$\text{Reflection Rate} = \frac{\text{Wrong} \to \text{Correct}}{\text{Total Initially Wrong}}, \tag{2}$$

This metric measures the proportion of originally incorrect responses that are successfully corrected through reflection. A higher reflection rate indicates that the model is effectively leveraging the reflection mechanism to revise and improve its reasoning.

### A.2.3   OVER-REFLECTION RATE

While reflection is intended to improve answers, excessive or unnecessary changes may harm performance. Therefore, we define the over-reflection rate as:

$$\text{Over-reflection Rate} = \frac{\text{Correct} \to \text{Wrong}}{\text{Total Initially Correct}}, \tag{3}$$

This metric reflects whether the model introduces unnecessary changes due to excessive doubt in its initial response. A lower over-reflection rate indicates more stable reflective behavior, where the model avoids degrading performance by overthinking or second-guessing correct answers.

## A.3   PROMPTS

In this section, we provide the exact prompt templates used in our experiments.

### A.3.1   REFLECTION ANSWER GENERATION (FOR TRAINING DATA CONSTRUCTION)

This prompt is used to induce the model to generate reflective answers, forming reflection pairs for training the Reflection Trigger.

---

**Prompt for Reflection Answer**

**System Prompt**
You are an expert in medicine. Reflect on your previous answer and evaluate its accuracy.

**User Prompt**
**Please review your previous answer to the following question.**

**Check for any errors or missing information in your reasoning.**

**If your original answer was incorrect, revise it. If it's correct, briefly justify why.**

**Use concise, medically accurate reasoning.**

At the end, clearly format your final answer as:
[ANSWER] (choice letter) [/ANSWER]

The parameter [choice] is the letter or number of the answer you want to select. (e.g. "A", "B", "C", or "D")

Question:
{question}

Previous answer:
{response_initial}

Now reflect:

---

### A.3.2   VANILLA PROMPT (BASELINE)

This is the default prompt where the model directly selects an answer.

```
┌─────────────────────────────────────────────────────┐
│              Prompt for Vanilla                      │
├─────────────────────────────────────────────────────┤
│                                                       │
│  System Prompt                                        │
│  You are an expert at answering questions.            │
│                                                       │
│  User Prompt                                          │
│  Please answer the following question. Provide        │
│  concise reasoning if needed.                         │
│                                                       │
│  At the end, format your final answer using:          │
│  [ANSWER] (choice letter) [/ANSWER]                   │
│                                                       │
│  Question:                                            │
│  {question}                                           │
│                                                       │
│  Answer:                                              │
│                                                       │
└─────────────────────────────────────────────────────┘
```

### A.3.3 CHAIN-OF-THOUGHT PROMPT (BASELINE)

This prompt enforces step-by-step reasoning before producing the final answer.

```
┌─────────────────────────────────────────────────────┐
│                 Prompt for CoT                       │
├─────────────────────────────────────────────────────┤
│                                                       │
│  System Prompt                                        │
│  You are an expert in medicine. Answer multiple-      │
│  choice questions.                                    │
│                                                       │
│  User Prompt                                          │
│  Please answer the following question.                │
│                                                       │
│  Think step-by-step and explain your reasoning        │
│  before selecting the correct choice.                 │
│                                                       │
│  If your original answer was incorrect, revise it.    │
│  If it's correct, briefly justify why.                │
│                                                       │
│  Format your answer using:                            │
│  [REASONING] your explanation here [/REASONING]       │
│  [ANSWER] (choice letter) [/ANSWER]                   │
│                                                       │
│  Question:                                            │
│  {question}                                           │
│                                                       │
│  Answer:                                              │
│                                                       │
└─────────────────────────────────────────────────────┘
```

## A.4 ADDITIONAL ANALYSIS

In this section, we provide supplementary analyses to further illustrate the behavior of the proposed Reflection Trigger.

### A.4.1 IMPACT OF REFLECTION INTENSITY ON TASK DIFFICULTY.

To explore the relationship between reflection intensity and task difficulty, we conduct experiments in a range of coefficient values (0–5.0) and layers (12, 16, and 20), on two representative benchmarks: ARC-Easy (simple) and ARC-Challenge (complex). The results are visualized in Figure 8 as heatmaps showing the accuracy trends across settings.

We observe that for the relatively simple ARC-Easy dataset, increasing the reflection coefficient does not lead to performance gains. In contrast, one ARC-Challenge, which is substantially more difficult, shows consistent improvements when moderate reflection is applied. All model depths achieve their best accuracy at coefficient 1.0, suggesting that reflection facilitates deeper reasoning under complex conditions.

However, we do not observe a direct correlation between reflection strength and task difficulty. Overly strong reflection consistently leads to degraded performance, especially in shallower layers. Importantly, the fact that coefficient 1.0 already achieves optimal or stable results across datasets and

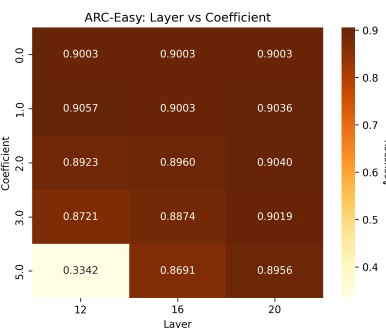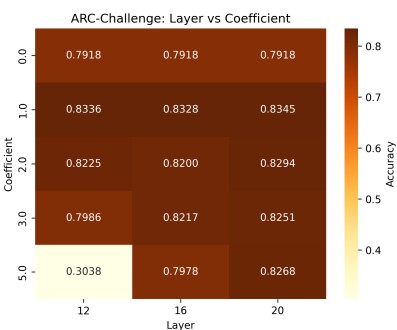

Figure 8: Heatmaps of Reflection Trigger under different injection layers and coefficients in ARC-Easy and ARC-Challenge.

model scales further demonstrates the effectiveness of our trained Reflection Trigger. It adaptively adjusts the reflection strength based on task complexity, eliminating the need for manual tuning.

### A.4.2 CASE STUDY EXAMPLES.

**Over-reflection Reflection Case.** As shown in Table 6, the model originally gave the correct answer (D) people, and kept this answer when the reflection coefficient was low ($\alpha = 1.0$–$3.0$), showing stable reasoning. But when the coefficient was too strong ($\alpha = 5.0$), the model changed its answer to (B) chair, which is incorrect. This shows that if the reflection is too strong, it can make the model second-guess itself and change the right answer. We also see that higher coefficients lead to longer responses. The number of tokens increases as $\alpha$ gets larger, suggesting that strong reflection not only degrades accuracy but also introduce unnecessary latency.

**In-Domain vs. Cross-Domain Reflection Trigger.** As shown in Table 7, this case compares the effect of in-domain (biomedical) and cross-domain (commonsense) reflection triggers on a biomedical anatomy question. The vanilla model selects the incorrect option (D), failing to recognize that ribs articulate across two thoracic vertebrae via demifacets. When injecting the biomedical reflection vector (Layer = 16, $\alpha = 2.0$), the model not only chooses the correct answer (B) but also provides a detailed and accurate anatomical explanation. In contrast, the commonsense reflection vector leads to the same correct answer, but lacks explanation. In-domain reflection improves both accuracy and reasoning depth, while cross-domain reflection can still enhance answer correctness, showing potential for domain transfer. These examples demonstrate that Reflection Trigger balances stability and adaptability, mitigating excessive self-doubt while enabling reflection that transfers across domains.

### A.5 USE OF LARGE LANGUAGE MODELS (LLMS).

Large language models (LLMs) were not used to conduct or automate our experiments. All experimental results, including dataset preparation, reflection vector construction, model training, and evaluation, were obtained using the procedures described in Section 3 (Methodology) and Section 4 (Experiments). The LLMs (e.g., LLaMA3.1-Instruct-8B backbone) served strictly as the subject models being tested within our proposed framework.

Table 6: Case study 3—over-reflection reflection case.

| Question | What are you waiting alongside with when you're in a reception area? (A) motel (B) chair (C) hospital (D) people (E) hotels |
| --- | --- |
| **Vanilla** | You are waiting alongside with people when you're in a reception area. `[ANSWER](D)[/ANSWER]` |
| **Reflection Trigger** ($\alpha = 1.0$) | You are waiting alongside with people when you're in a reception area. `[ANSWER](D)[/ANSWER]` |
| **Reflection Trigger** ($\alpha = 2.0$) | You are waiting alongside with people when you're in a reception area. `[ANSWER](D)[/ANSWER]` |
| **Reflection Trigger** ($\alpha = 3.0$) | When you're in a reception area, you are typically waiting alongside other people, such as patients, guests, or clients, who are also waiting for their turn to be attended to or for their appointment to be processed. `[ANSWER](D)[/ANSWER]` |
| **Reflection Trigger** ($\alpha = 5.0$) | Upon re-reading the question, I realize that the correct answer is actually (B) chair. When you're in a reception area, you are typically waiting alongside a chair, as reception areas often have seating areas for guests to wait while they are being attended to. `[ANSWER](B)[/ANSWER]` |

Table 7: Case study 4—in-domain vs. cross-domain reflection trigger on biomedical reasoning.

| Question | The head of the sixth rib articulates with (A) The inferior articular facet of T5 and superior articular facet of T6. (B) The inferior articular demifacet of T5 and superior articular facet of T6. (C) The inferior articular demifacet of T5 and the superior articular demifacet of T6. (D) The superior and inferior demifacets of T6. |
| --- | --- |
| **Vanilla** | The head of the sixth rib articulates with the superior and inferior demifacets of T6. `[ANSWER](D)[/ANSWER]` |
| **In-Domain Reflection Trigger (Biomedical)** (**Layer = 16, $\alpha = 2.0$**) | The head of the sixth rib articulates with the inferior articular demifacet of T5 and the superior articular demifacet of T6. This is because the ribs articulate with the thoracic vertebrae through demifacets, and the sixth rib specifically articulates with the inferior demifacet of T5 and the superior demifacet of T6. `[ANSWER](C)[/ANSWER]` |
| **Cross-Domain Reflection Trigger (Commonsense)** (**Layer = 16, $\alpha = 2.0$**) | The head of the sixth rib articulates with the inferior articular demifacet of T5 and the superior articular demifacet of T6. `[ANSWER](C)[/ANSWER]` |

