# OpenReview forum: "Reflection Trigger: Latent Self-Correction for Question Answering by Steering Vector Injection"
_ICLR.cc/2026/Conference — Submitted to ICLR 2026_

### Official Review · Reviewer_1eQh · 2025-10-24

**Soundness:** 3
**Presentation:** 3
**Contribution:** 2
**Rating:** 2
**Confidence:** 3

**Summary:**

The paper proposes a vector-based mechanism that dynamically injects a vector into LLMs during inference, which recasts prompt-induced reflection as a learnable latent direction and demonstrates that activation addition, predicted from the input, can improve reasoning accuracy and reduce over-reflection.

**Strengths:**

1. Predicting a per-input steering vector and injecting it once is conceptually clean and easy to implement.
2. The proposed method improves model reasoning accuracy while reducing over-reflection errors.

**Weaknesses:**

1. It’s unclear which token representations are compared (CLS, final token, mean pooling) or how sequences of different lengths are aligned.
2. The ground-truth reflection vector is computed from reflection-prompted answers, and training keeps only cases where reflection yields a correct final answer. This filtering injects selection bias that may overstate the learned reflection quality. It’s closer to learning to approximate successful reflection than a good reflection direction.
3. All experiments use only Llama-3.1-Instruct-8B.
4. Reflection pairs are produced by the same backbone model later evaluated; if dataset splits are not airtight, the trigger module might indirectly fit test distribution artifacts.
5. The Wait and CoT baselines are underpowered (no self-consistency, limited tuning), and there’s no comparison to existing activation-steering baselines (e.g., ActAdd, contrastive vector addition). LoRA baselines are included but trained differently, so comparisons are not strictly fair.
6. For the ablation study, the paper omits comparisons such as a global, input-agnostic vector vs. a per-input predictor, or random/shuffled vectors.
7. Injected vectors can push activations in poorly understood ways, the paper notes failure under large $\alpha$ but lacks safety checks such as semantic drift or adversarial robustness

**Questions:**

The reflection and over-reflection rates are defined in terms of changes from the initial answer. How do you implement for methods without multi-turn interaction?

---

### Official Review · Reviewer_n9dN · 2025-10-29

**Soundness:** 2
**Presentation:** 2
**Contribution:** 1
**Rating:** 2
**Confidence:** 5

**Summary:**

This paper proposes a method to control LLMs' reflection tendencies during inference time, and thereby improves their accuracy. The method works by learning an input-conditioned "reflection vector" during training, and then adding the reflection vector into an LLM's intermediate hidden state scaled by a tunable coefficient to control reflection intensity. Experiments show that the proposed method improves accuracy and reduce over-reflection.

**Strengths:**

1. Steering LLMs' behaviors through latent representations is an interesting direction and has been shown effective in tasks such as controlling style or sentiment. Extending this method to induce reflective reasoning is a reasonable direction to explore.

2. Empirical evidence shows that Reflection Trigger improves accuracy and reduce over-reflection.

**Weaknesses:**

1. There is an important missing baseline, which is training LLMs with reinforcement learning with verifiable rewards (RLVR). It has been widely known that RLVR can improve LLMs' performance on reasoning tasks dramatically, and LLMs also learn "reflection behaviors" such as answer verification after RL training. For example, an existing work [1] apply RLVR (i.e., the work called it minimalist rule-based RL) on medical datasets and achieved accuracies of 76.19% on MedQA and 64.47% on MedMCQA with RL-trained Llama-3.1-8B-Instruct, which is the same backbone model used in learning Reflection Trigger. However, Reflection Trigger's performance only improves to 64.10% on MedQA and 56.44% on MedMCQA. This makes me wonder whether it is necessary to adopt this, in my opinion, more complicated approach rather than plain RLVR.


2. Following the previous point, Reflection Trigger requires a BERT encoder to learn the reflection vector. While RLVR with the GRPO objective only requires the backbone LLM itself. Considering RLVR's simpler training setup and its much superior performance, I currently don't see why this is not included as a baseline and the advantage of Reflection Trigger over RLVR.

Overall, my biggest concern is that the baselines are too weak to demonstrate the contributions of the proposed method in terms of improving reasoning accuracy, which is one of the authors' major claim.

Reference:

[1] Liu, C., Wang, H., Pan, J., Wan, Z., Dai, Y., Lin, F., ... & Arcucci, R. (2025). Beyond distillation: Pushing the limits of medical llm reasoning with minimalist rule-based rl. arXiv preprint arXiv:2505.17952.

**Questions:**

1. I suggest that the authors should compare Reflection Trigger with RLVR-based methods thoroughlly. Except for accuracy, it would also be nice to compare the efficiency trade-off like generated tokens.

2. How do you choose the final hyperparameters used during testing? For example, how do you determine which intermediate layer (m) and the value of injection coefficient (alpha)? Did you use a validation set? It is not explicitly mention in the paper.

---

### Official Review · Reviewer_hx8k · 2025-10-30

**Soundness:** 2
**Presentation:** 2
**Contribution:** 2
**Rating:** 4
**Confidence:** 4

**Summary:**

This paper aims to address the problems of over-reflection, high latency, and instability in existing prompt-based reflection mechanisms in LLMs. The authors propose a novel method named "Reflection Trigger," which injects a dynamically learned, input-specific "reflection vector" into the intermediate layers of a frozen LLM during inference, without requiring modification of the model parameters. This reflection vector is learned by a separate, lightweight module (a BERT encoder and a linear layer), and its training objective is to predict the difference in hidden states when the model generates responses with versus without a reflection prompt. Experiments on biomedical and commonsense question-answering datasets demonstrate that this method effectively improves model accuracy while significantly reducing the over-reflection rate and the number of generated tokens.

**Strengths:**

1. Unlike strategies such as the "Wait" prompt that require multi-turn interaction, the proposed method can guide the model to reflect within a single forward pass, reducing inference latency and computational overhead.
2. The method allows for some degree of control over the intensity of reflection, enhancing its granularity.
3. The paper includes extensive experiments to demonstrate the effectiveness of the proposed method.

**Weaknesses:**

1.The primary concern is that the method's main improvements may stem from the construction of a high-quality reflection dataset and the use of this dataset to train an implicit reflection prompt injector. However, the paper lacks a comparison of this injector's superiority over other advanced knowledge injection methods on the same dataset. This limits the novelty and insightfulness of the proposed method.
2.The reflection baselines used for comparison are not sufficiently strong, potentially leading to an unfair comparison. Furthermore, when validating the method's cross-domain performance, the authors do not compare it against other methods, making it difficult to prove the proposed method's superior robustness.
3.Section 3.2.2 Model Training and the general training details lack sufficient information. For example, the loss function is not defined; it is unclear whether the BERT encoder is frozen or fine-tuned during the Trigger's training; and there is an inconsistency between the inputs to the Trigger shown in Figure 5 and Figure 3, which might be a typo.
4.The paper only experiments on LLaMA-3.1 and lacks experiments and comparisons on other models known for stronger CoT capabilities, such as Qwen.

**Questions:**

1. Why did you choose to model the difference (RA hidden state - IA hidden state) rather than modeling the RA hidden state directly? What impact does this choice have?
2. Is the extracted hidden state the state of the last token, an average pooling of the states of all answer tokens, or derived from another aggregation method?
3. How robust is the proposed method to variations in test-time prompts? Furthermore, what happens if a test prompt conflicts with the reflection instructions used to create the training data? What is the impact of this method on multi-turn dialogues?

---

### Official Review · Reviewer_CsXy · 2025-11-01

**Soundness:** 2
**Presentation:** 2
**Contribution:** 3
**Rating:** 4
**Confidence:** 4

**Summary:**

The motivation of this paper is that large language models (LLMs) can perform reasoning tasks, but achieving stable reflective reasoning remains a major challenge. Existing prompt engineering approaches, such as Chain-of-Thought (CoT) or the “What,” prompt, can generate reasoning steps but suffer from several drawbacks, including overthinking, high sensitivity to prompt formulation, and instability. Moreover, these methods do not allow fine-grained control over the intensity of reflection.

To address these limitations, the authors propose a vector-guided reflection control mechanism called the Reflection Trigger. As illustrated in Figure 2, this mechanism injects a learned vector into the intermediate layers of the model during inference, enabling dynamic adjustment of reflective tendencies across different questions.

The reflection (or steering) vector is generated by adding a layer on top of a BERT encoder output, matching the dimensionality of the hidden representations of the target LLM at a given layer. The input question is fed to the BERT encoder. For training, the authors use a prompting scheme where the LLM is asked to reconsider its previous answer; they then compute the difference between the LLM’s hidden states for the initial prompt and for the “reflective” prompt at a given layer, and use this as supervision.

The experiments aim to answer three research questions:
- RQ1: Can Reflection Trigger improve reasoning accuracy across domains?
- RQ2: Does Reflection Trigger reduce over-reflection while enabling effective self-correction?
- RQ3: How does Reflection Trigger compare with prompt-based reflection methods in terms of over-reflection control and efficiency?

The method is compared against several baselines: (1) the vanilla model without any intervention, (2) LoRA, (3) CoT prompting, and (4) the “Wait,” prompt. Experiments are conducted on five datasets spanning two domains: Biomedical Reasoning and Commonsense Reasoning. The evaluation metrics include accuracy, as well as flip rates—the rates at which correct answers become incorrect (and vice versa) after reflection.

Results show that the proposed Reflection Trigger improves reasoning accuracy and effectively reduces over-reflection compared to existing prompt-based reflection methods.

**Strengths:**

- The authors propose a novel approach to compute a steering vector that induces a reflective phase in the LLM, allowing it to potentially revise its answers. The idea of learning a model to automatically generate this vector via an encoder is interesting and well-motivated.
- The authors demonstrate transferability of the learned reflection vector across two domains: Biomedical Reasoning and Commonsense Reasoning. The proposed model allows for controlled adjustment of the intensity with which the LLM is pushed toward reflective reasoning.
- The proposed model achieves higher average accuracy than the compared baselines while generating fewer tokens and maintaining a lower rate of over-reflection.

**Weaknesses:**

- Key parts of the approach are presented in the appendix, whereas they would be better placed in the main body of the paper to ensure clarity and accessibility.
- The method is evaluated on only a single LLM and a single type of encoder. It is unclear whether the approach would generalize to LLMs of different sizes or architectures.
- The effectiveness of learning the reflection vectors via the BERT output layer is not studied independently. It remains unclear how one could evaluate whether the model generates the intended vectors beyond the overall accuracy of the full model.

**Questions:**

Discussion:
While the paper proposes an interesting and novel method for inducing reflective reasoning in LLMs, several points merit further clarification and discussion:
- RQ1 (Reasoning Accuracy): The accuracy results of the proposed method compared to CoT are very close. It is unclear whether the Reflection Trigger method is significantly better in terms of reasoning accuracy.
- RQ2 (Over-reflection and Self-Correction): The authors highlight an improved over-reflection rate (the rate of flipping a correct answer to an incorrect one) compared to the “Wait,” prompt. However, they do not discuss the corresponding reflection rate (the rate of flipping an incorrect answer to a correct one). Is the improvement primarily due to the Reflection Trigger achieving higher overall accuracy, rather than better control over reflection?
- RQ3 (Comparison with Prompt-based Reflection Methods): On line 419, the paper mentions training with 0% of the training data, but it is unclear what this means in practice.
- Placement of Content: Several sections, including Section 5, could arguably be moved to the appendix without losing clarity. Additionally, key parts of the method are presented in the appendix rather than the main text, which reduces accessibility.

Minor Comments (not affecting overall score):
- The example in Figure 2 appears physically inaccurate: water in cold air is not vapor but supercooled droplets.
- Line 289, Section A.3.1 refers to the prompt used to generate the Reflection Answer rather than the data filtering procedure for reflection training. Since the selection procedure is straightforward, a brief explanation in the main text would improve clarity.
- Definitions of over-reflection rate and reflection rate are provided only in the appendix, but they could be quickly summarized in the main text.

---

### Meta-Review · Area_Chair_vekW · 2025-12-26

**Summary:**

Reviewer n9dN points out that the proposed method significantly underperforms compared to existing strong baselines like RLVR on the same benchmarks (64% vs. 76%), undermining the motivation for the proposed complex vector injection mechanism. The evaluation is considered too narrow, relying solely on a single model (LLaMA-3.1) and utilizing underpowered CoT baselines, which makes the claimed improvements unconvincing. Serious concerns regarding methodological flaws, such as selection bias in training data construction and missing technical details (e.g., token alignment, loss formulation), further justify the rejection.

**Reviewer Concerns:**

No rebuttal was submitted.

**Reviewer Scores:**

No rebuttal was submitted.

---

### Decision · Program_Chairs · 2026-01-26

Reject